# Impact of Molecular Biology in Diagnosis, Prognosis, and Therapeutic Management of *BCR::ABL1*-Negative Myeloproliferative Neoplasm

**DOI:** 10.3390/cells12010105

**Published:** 2022-12-27

**Authors:** Norman Abbou, Pauline Piazzola, Jean Gabert, Vincent Ernest, Robin Arcani, Anne-Laure Couderc, Antoine Tichadou, Pauline Roche, Laure Farnault, Julien Colle, L’houcine Ouafik, Pierre Morange, Régis Costello, Geoffroy Venton

**Affiliations:** 1Molecular Biology Laboratory, North University Hospital, 13015 Marseille, France; 2INSERM, INRAE, C2VN, Aix-Marseille University, 13005 Marseille, France; 3Hematology and Cellular Therapy Department, Conception University Hospital, 13005 Marseille, France; 4Hematology Laboratory, Timone University Hospital, 13005 Marseille, France; 5Department of Internal Medicine, Timone University Hospital, 13005 Marseille, France; 6Department of Geriatrics, South University Hospital, 13005 Marseille, France; 7CNRS, INP, Institute of Neurophysiopathol, Aix-Marseille Université, 13005 Marseille, France; 8APHM, CHU Nord, Service d’Onco-Biologie, Aix-Marseille Université, 13005 Marseille, France; 9TAGC, INSERM, UMR1090, Aix-Marseille University, 13005 Marseille, France

**Keywords:** myeloproliferative neoplasms, next-generation sequencing, driver mutations, additional somatic mutations

## Abstract

*BCR::ABL1*-negative myeloproliferative neoplasms (MPNs) include three major subgroups—polycythemia vera (PV), essential thrombocythemia (ET), and primary myelofibrosis (PMF)—which are characterized by aberrant hematopoietic proliferation with an increased risk of leukemic transformation. Besides the driver mutations, which are *JAK2, CALR, and MPL*, more than twenty additional mutations have been identified through the use of next-generation sequencing (NGS), which can be involved with pathways that regulate epigenetic modifications, RNA splicing, or DNA repair. The aim of this short review is to highlight the impact of molecular biology on the diagnosis, prognosis, and therapeutic management of patients with PV, ET, and PMF.

## 1. Introduction

*BCR::ABL1*-negative myeloproliferative neoplasms (MPNs) are a group of stem cell disorders characterized by aberrant hematopoietic proliferation with an increased risk of leukemic transformation. BCR::ABL1-negative MPNs include three major subgroups: polycythemia vera (PV), essential thrombocythemia (ET)**,** and primary myelofibrosis (PMF) [1]. A major advancement in our understanding of the genetic landscape of MPNs has been made following the discovery of the *JAK2 V617F* somatic mutation in 2005. Most of the MPNs have one or more somatic mutations that activate a signaling pathway, conferring a proliferative advantage to the neoplastic cells. Regarding PV, ET, and PMF, the main “driver” mutations affect the *JAK2, CALR, and MPL* genes, activate the JAK–STAT signaling pathway, and induce a proliferation and antiapoptotic effect in cells. Besides these driver mutations, additional mutations may coexist in pathways that regulate epigenetic modifications, RNA splicing, or DNA repair [2,3,4]. In 2020 and 2021, Loscocco et al. and Ross Dm et al., respectively, listed twenty additional mutations (*ASXL1, CBL*, *DNMT3A, EZH2, IDH1/2, RUNX1, SETBP1, SF3B1, SRSF2, TET2, ZRSR2, LNK/SH2B3, NRAS/KRAS, PTPN11, TP53, and U2AF1*) that they identified using next-generation sequencing (NGS) analyses. These mutations occur between 0.5 and 10% in patients with MPN [5,6]. Table 1 summarizes all additional mutations described in patients with PV, ET, and PMF with their respective types and frequencies. The aim of this short review is to highlight the impact of NGS on the diagnostic, prognostic, and therapeutic management of patients with PV, ET, and PMF in 2022. The *SF3B*1 mutation (associated with myelodysplastic syndrome) is not within the scope of this review.

## 2. Impact of Mutational Profile on PV, ET, and PMF Diagnosis

MPNs are characterized by a constitutive activation of the JAK–STAT pathway. Mutations in *JAK2, CALR, and MPL* are referred to as “driver mutations” because they lead to the determination of the MPN phenotype. In addition, based on the WHO’s 2016 revision to the classification of myeloid neoplasms and acute leukemia, these mutations, together with the clinical, biological, and histological features, allow for a determination of the diagnosis [1]. Even though these three mutations cannot usually coexist, a few cases have been reported where patients turned out to have rare, mutated variations of JAK2 and MPL, particularly in patients initially ranked “without driver mutations” or with a low allelic burden for driver mutations in ET [5,34,35,36]. However, a subset (10–15%) of patients with MPN, called triple-negative (TN) patients, do not present any of these driver mutations but may harbor others that might be associated with a shorter overall survival (OS) and/or leukemia-free survival (LFS) [7,37,38].

### 2.1. Somatic Myeloid NGS Panel and Erythrocytosis Exploration

As described in the RESPONSE clinical trial, almost 99% of PV cases are associated with somatic mutations in *JAK2* (97.3% in JAK2 V617F with a mean variant allele frequency (VAF) of approximately 84% and 1.3% in exon 12) and are hypercellular with trilineage hyperplasia (panmyelosis) with evidence of an increased red cell mass [39]. Moreover, many clinical situations can occur and should be considered. We suppose in these clinical cases that a secondary cause of polyglobulia is not present:*Clinical situation No. 1:* The patient has an absolute erythrocytosis according to the WHO 2016 PV diagnostic criteria (hemoglobin levels > 18.5 g/dL in men (hematocrit > 55.5%) or >16.5 g/dL in women (hematocrit > 49.5%)), and the *JAK2* mutation with a high VAF is present. If the EPO level is suppressed (as in 80% of PV), a bone marrow (BM) biopsy is not required [1,40]. A PV diagnosis is confirmed.*Clinical situation No. 2:* The patient has an absolute erythrocytosis according to the WHO 2016 PV diagnostic criteria without the *JAK2* mutation (or with a low VAF). Even if the EPO level is suppressed, a BM biopsy is required to highlight BM hypercellularity (present in 90% of PV) and to confirm a PV diagnosis [41].*Clinical situation No. 3:* The patient has an erythrocytosis according to the WHO 2016 classification (HGB > 16.5 g/dL (men) and >16 g/dL (women), HCT > 49% (men) and >48% (women) or increased red cell mass) without meeting the absolute erythrocytosis criteria. The *JAK2* mutation is present at a high ratio, and a BM biopsy highlights BM hypercellularity. A PV diagnosis is confirmed.

In these three clinical situations outlined above, a PV diagnosis is not a clinical issue. However, as described above, 1% of patients with PV are *JAK2* negative, as the EPO level is not suppressed in 20% of cases, and the BM does not show panmyelosis in 10% of JAK2-positive erythrocytosis. In these rare cases, in which PV is suspected and/or the WHO diagnostic criteria are met despite the absence of a detectable *JAK2* mutation, a somatic myeloid NGS panel can help with PV diagnosis in addition to research on noncanonical nucleotide changes leading to V617F or compound mutations in exon 14 [42,43]. In association with concordant BM features, NGS can provide evidence of clonal hematopoiesis, notably by detecting alternative activating JAK–STAT pathway driver mutations, such as *LNK/SH2B3* or nondriver mutations such as *TET2 or DNMT3A* [44].

When the myeloid NGS panel and *JAK2* are both negative, the detection of mutations in other genes via germline NGS panels may be useful to diagnose hereditary erythrocytosis [44]. In the majority of laboratories, these genes are typically not included in myeloid NGS panels, but there is some overlap between the germline and somatic candidate gene lists [6]. In the case of *JAK2*-negative erythrocytosis without panmyelosis in the BM and the presence of a nondriver mutation found through NGS, clonal hematopoiesis of indeterminate potential (CHIP) must be discussed, especially when genes such as *DNMT3A, ASXL1, or TET2* are involved [45]. If the somatic and germline myeloid NGS panel are also negative, idiopathic or secondary erythrocytosis should be considered, especially if the EPO level is not suppressed (normal or increased). The different erythrocytosis and PV diagnostic approaches are summarized in Figure 1.

### 2.2. Impact of Mutational Status on ET and PMF Diagnosis

ET is characterized by thrombocytosis and megakaryocytic hyperplasia. Because there are many causes of reactive thrombocytosis, the use of driver or nondriver mutations to provide the proof of clonality is essential to make an ET diagnosis. Driver mutations in *JAK2, CALR, and MPL* are detected in 60–65%, 20–25%, and 5% of patients with ET, respectively [2,30]. Therefore, between 10 and 20% of patients with ET are TN. Outside of some notable cases reported, *JAK2, CALR, and MPL* driver mutations are mutually exclusive in ET. Regarding the *JAK2* V617F mutation in ET, the median VAF is lower than in PV or PMF. In ET, the median VAF is approximately 10 to 20% and rarely exceeds 50% compared to a median VAF frequency above 50% in PV and PMF [46,47]. With regard to *CALR*, more than 50 mutations have been described for which *CALR* type 1 (52 base-pair deletions) and type 2 (5 base-pair insertions) represent 80% of cases [48]. In the case of thrombocytosis (with a platelet count > 450.109/L) with no identified driver mutation, no evidence for reactive thrombocytosis, and a strong suspicion of ET, a BM biopsy should be the first intention exam before the myeloid NGS panel. According to the WHO 2016 ET diagnostic criteria, a BM biopsy showing proliferation mainly in the megakaryocytic lineage with increased numbers of enlarged, mature megakaryocytes with hyperlobulated nuclei is a major diagnostic criterion, whereas the presence of a clonal marker found through NGS, for example, is a minor criterion [1]. In the absence of any MPN features in the BM, an ET diagnosis cannot be retained. On the contrary, in the presence of concordant BM, a noncanonical mutation in *CALR, JAK2, or MPL* and/or nondriver additional mutations detected through NGS can help to make a definitive diagnosis of ET and can help provide information on the patient’s prognosis (see below).

Regarding PMF, the distribution of driver mutations is similar to that in ET, as it is also 10% of TN patients [49]. As with patients with ET, NGS can detect noncanonical mutations in *CALR, JAK2, or MPL* and/or nondriver additional mutations. Contrary to ET, if the BM biopsy shows the presence of megakaryocytic proliferation and atypia accompanied by either reticulin and/or collagen fibrosis grade 2 or 3 and if reactive causes can be excluded, then a clonality marker is not strictly necessary to make a PMF diagnosis. The WHO 2016 MPN diagnostic criteria revision identified a fourth and independent entity: prefibrotic myelofibrosis (pre-PMF) [1]. The distinction between ET and pre-PMF is not always simple and is theoretically based on morphological criteria with variable reproducibility [50,51]. There are also more robust, minor pre-PMF diagnostic criteria, such as anemia, palpable splenomegaly, leukocytosis, or a raised lactate dehydrogenase level [1]. These minor criteria are classically absent in ET, which instead has criteria such as a high VAF of JAK2V617F above 50% and/or the presence of additional nondriver mutations found through NGS, such as *NRAS, ZRSR2, U2AF1,* and *SRSF2* (see Table 1) [46]. The different ET and PMF diagnostic approaches are summarized in Figure 2.

## 3. Impact of Somatic Myeloid NGS Panel in MPN Prognosis and Treatment Management

In 2016, Tefferi et al. performed deep NGS in 133 patients with PV and 183 patients with ET. A little more than half of the patients with PV and patients with ET presented additional nondriver mutations. The most frequent were *TET2* and *ASXL1*. The *TET2* mutation occurs in 15 and 10% of patients with PV and patients with ET, respectively (Table 1). In patients with both *JAK2 V617F* and *TET2* mutations, the order in which the *JAK2* and *TET2* mutations were acquired influenced the clinical features, the response to targeted therapy, the biology of stem and progenitor cells, and the clonal evolution. Indeed, as compared with patients in whom the *TET2* mutation was acquired first, patients in whom the *JAK2* mutation was acquired first showed a larger probability of presenting with PV than ET, an increased risk of thrombosis, and an increased in vitro sensitivity of *JAK2*-mutant progenitors to ruxolitinib. Mutation order influenced the proliferative response to *JAK2* V617F and the capacity of double-mutant hematopoietic cells and progenitor cells to generate colony-forming cells [8]. High-risk mutations (HRM = adverse prognostic) included *ASXL1, SRSF2,* and *IDH2* in patients with PV, and *SH2B3, SF3B1, U2AF1, TP53, IDH2,* and *EZH2* in patients with ET. These HRMs may be combined in 15% of patients with PV and/or ET. HRMs were associated with inferior OS in patients with PV (median OS, 7.7 vs. 16.9 years) and patients with ET (median OS, 9 vs. 22 years) [9].

The implementation of NGS techniques allows for the identification of additional mutations in patients that influence survival and the risk of transformation into myelofibrosis. A prognostic score called the mutation-enhanced international prognostic model in PV and ET (MIPSS-PV and MIPSS-ET), based on an age > 67 years, leukocytosis (≥15 × 109/L), an abnormal karyotype, and the presence of additional poor prognostic mutations (*SRSF2* in *PV, SRSF2, SF3B1*, *U2AF1*, and *TP53* in ET) allowed for the establishment of a classification of the survival of patients. The classification of low risk, intermediate risk, and high risk with corresponding median survivals of 24, 13.1, and 3.2 years, respectively, can help modulate the management, prognosis, and therapy for the patient [52].

As for patients with PV and patients with ET, 182 patients with PMF were screened with a somatic myeloid NGS panel. Additional mutations were detected in 81% of patients with PMF. The most frequent mutations were *ASXL1* (36%), *TET2* (18%), *SRSF2* (18%), and *U2AF1* (16%) (Table 1); furthermore, 35%, 26%, 10%, and 9% of the patients harbored 1, 2, 3, and 4 or more such variants/mutations, respectively. HRMs in patients with PMF, with a pejorative impact on OS or LFS, included *ASXL1, SRSF2, CBL, KIT, RUNX1, SH2B3, and CEBPA*; their combined prevalence was 56%. HRMs were associated with an inferior OS (median, 3.6 vs. 8.5 years; *p* < 0.001) and LFS (7-year risk, 25% vs. 4%; *p* < 0.001), and the effect on survival was independent of both the Dynamic International Prognostic Scoring System Plus (DIPSS+) and *JAK2/CALR/MPL* mutational status, especially for *ASXL1* [7,13,18].

In 2014, 570 patients with PMF were recruited to build a molecular prognostic model based on *CALR and ASXL1* mutations. OS was the longest in *CALR (+) ASXL1 (-)* patients (median OS of 10.4 years) and shortest in CALR (-) ASXL1 (+) patients (median OS of 2.3 years). *CALR (+) ASXL1 (+)* and *CALR (-) ASXL1 (-)* patients had similar survival rates and were grouped together in an intermediate-risk category (median OS of 5.8 years). The *CALR/ASXL1*-mutation-based prognostic model was independent of the DIPSS+ (*p* < 0.0001) and effective at identifying DIPSS+ low/intermediate-1-risk patients with a shorter (median OS of 4 years) or longer (median OS of 20 years) survival and high/intermediate-2-risk patients with a shorter (median OS of 2.3 years) survival. A multivariable analysis distinguished the *CALR (-) ASXL1 (+)* mutational status as the most significant risk factor for survival in older patients (age > 65 years) and/or those with an unfavorable karyotype [14]. More recently, a risk stratification based on *ASXL1* and/or *SRSF2* was proposed for patients with PMF and a low or intermediate-1 risk in the DIPSS+. In this specific subgroup, patients with PMF and *ASXL1* and/or *SRSF2* have a significantly lower OS than patients with PMF without these two mutations (median OS of approximately 18 years vs. 5 years, *p* < 0.0001) [15].

In 2017, Newberry et al., based on patients with MF enrolled in a ruxolitinib discontinuation phase 1/2 study, reported that patients with three or more additional mutations (not only HRMs) are less likely to respond to ruxolitinib. After a median follow-up of 79 months, 86 patients had discontinued ruxolitinib (30 of whom died while undergoing therapy). The median follow-up after ruxolitinib discontinuation for the remaining 56 patients was 32 months with a median survival after discontinuation of 14 months. Platelets <260 × 109/L at the start of therapy or <100 × 109/L at the time of discontinuation were associated with a shorter survival after discontinuation. Of the 62 patients with molecular data at baseline and follow-up, 22 (35%) acquired a new mutation while receiving ruxolitinib (14 [61%] in *ASXL1*). Patients showing clonal evolution had a significantly shorter survival after discontinuation (6 vs. 16 months). Transfusion dependency was the only clinical variable associated with clonal evolution [53].

In 2018, Grinfeld et al. used NGS to sequence 69 myeloid cancer genes from a panel of 1887 patients to develop a genomic classification and prognostic models of MPNs [30]. They highlighted:Forty-five truncating mutations on the terminal exon of PPM1D, which is the eighth most frequent mutation;The mutation of the *MLL3* gene (nonsense or frameshift), which was detected in 20 patients and has already been described in patients with AML;Some noncanonical variants in 16 triple-negative patients with ET.

They, therefore, defined eight subgroups:(1)MPNs with *TP53* mutation (often associated with Ch17p aberration and 5q deletion);(2)MPNs with chromatin or spliceosome regulator mutations;(3)MPNs associated with *CALR* mutation;(4)MPNs associated with *MPL* mutation;(5)MPNs associated with *JAK2* or homozygous NFE2 mutation;(6)MPNs associated with heterozygous *JAK2* mutation;(7)MPNs with unknown driver mutation;(8)MPNs with another driver mutation.

According to the diagnostic groups mentioned above, Grinfeld et al. suggested a prognostic model based on the statistical results established by using a comparison with the JAK2 heterozygous subgroup (sixth subgroup in diagnostic classification). The most significant results are listed below [2]:-The first subgroup (MPNs with *TP53* mutation) was found to be associated with a higher risk of acute myeloid leukemia transformation and earlier death (HR 15.5; 95% CI, 7.5 to 31.4; *p* < 0.001).-The second subgroup (MPNs with chromatin or spliceosome regulator mutations) was exposed to a higher risk of transformation to myelofibrosis (HR 5.4; 95% CI, 2.7 to 11; *p* < 0.001) and had a shorter event-free survival (HR 2.6; 95% CI, 2.1 to 3.2; *p* < 0.001).-The patients in the fourth subgroup (MPNs associated with MPL mutation) with MF developed an increased risk of transformation to myeloid leukemia (HR 8.6; 95% CI, 1.4 to 49.1; *p* = 0.02).-The fifth group (MPNs associated with *JAK2* or homozygous *NFE2* mutation) showed a higher risk for myelofibrosis transformation (HR 3.0; 95% CI, 1.3 to 6.6; *p* = 0.007).-The eighth subgroup was associated with a good prognosis and only 0.5% and 1% of myelofibrosis and acute leukemia transformation, respectively (HR 0.56; 95% CI, 0.38 to 0.78; *p* = 0.005).

In 2020, Dunbar et al. worked on the leukemias secondary to myeloproliferative neo-plasms. They found some differences between post-MPN AML and de novo leukemia, which suggests different mechanisms in leukemogenesis [54]. For example, a higher rate of erythroblastic and megakaryoblastic leukemias are classically secondary to MPNs compared to the de novo leukemia. In addition, *FLT3, NPM1*, and *DNMT3A* are frequently absent in post-MPN AML (in contrast with de novo leukemia), whereas *IDH1/2, TET2, ASXL1, EZH 2,* and *SRSF2* are frequently present.

Furthermore, the “triple-negative” MPNs have an increased tendency to transform themselves into leukemia. Therefore, the study highlights some strong risk factors for the leukemic transformation of MPNs through:-Cytogenetic abnormalities, such as a monosomal karyotype, complex karyotype, or two sole abnormalities that include +8, -7/7q, i(17q), -5/5q-, and a chromosome 17p deletion.-Molecular abnormalities, such as *TP53, TET2, ASXL1, EZH2, SRSF2, IDH1/2, RUNX1,* and *U2AF1Q157*.

On the other hand, clonality studies have shown that the order of acquisition of mutations can change. Some earlier mutations in epigenetic genes, such as *ASXL1, TET2*, or *DNMT3A*, may lead to a driver mutation the second time, whereas mutations in *IDH1/2* generally occur after JAKV617F. When a *TET2* mutation occurs before a *JAK2* mutation, there is a higher risk of leukemic transformation.

At present, the mechanisms of how mutational order influences stem cells remain unclear. It would appear that some genetic or epigenetic mutations are required in addition to a driver mutation and/or other specific mutation to engender leukemic transformation. For instance, patients with *JAK2V617F/IDH1/2* mutations present a higher risk for leukemic transformation, and sometimes, this would involve the loss of the *JAK2V617F* mutation. This hypothesis is shared by Grinfeld et al. [2], who found that mutations in epigenetic regulators, splicing factors, and RAS signaling pathways are associated with a higher risk of progression to MF or AML. Further progress is still needed to identify patients at risk of progression.

The clinical and biological features of a PMF diagnosis, such as karyotypes and driver and/or nondriver mutations of PMF, alter a patient’s outcome. At diagnosis, 80% of patients with PMF harbor at least one HMR mutation, and NGS is the best way to detect these genetic changes [7]. In 2018, Tefferi et al. proposed a mutation- and karyotype-enhanced international prognostic scoring system for PMF (MIPSS70+) to better stratify the patient’s risk profile at diagnosis. The MIPSS70+ is based, notably, on the PMF HRMs previously described (*ASXL1, SRSF2, EZH2, IDH1, IDH2*, and *U2AF1)* [55]. Five risk groups were determined, very low, low, intermediate, high, and very high risk, for which the 10-year OS was 92%, 56%, 37%, 13%, and <5%, respectively. In the same way, Tefferi et al. proposed a genetically inspired prognostic scoring system exclusively based on genetic markers (GIPSS). Based on the analysis of 641 patients with PMF, a multivariable analysis identified a very high risk and unfavorable karyotype, the absence of the type 1/like *CALR* mutation, and the presence of the *ASXL1, SRSF2*, or *U2AF1Q157* mutation as inter-independent predictors of inferior survival. Four risk groups were determined: high, intermediate-2, intermediate-1, and low risk. The median OS determined with the GIPSS was 2 years for high risk, 4.2 years for intermediate-2 risk, 8 years for intermediate-1 risk, and 26.4 years for low risk. This score is a useful and simple tool, which requires only one karyotype and a few types of mutations. [49].

Allo-HSCT is the only curative therapy for patients with PMF when they are eligible. The median OS of any patient with PMF is approximately 5 years [56]. An optimal risk profile assessment at diagnosis is essential to select patients with PMF in whom allo-HSCT may be justified, especially for younger patients [57]. In 2016, Kroger et al. studied how molecular genetics may influence the outcome for patients with myelofibrosis after allo-HSCT. They screened 169 patients with PMF (*n* = 110), post-ET/PV MF (*n* = 46), and transformed MF (*n* = 13) for mutations on 16 frequently mutated genes. The most frequent mutation reported dealt with *JAK2V617F* (*n* = 101) followed by *ASXL1* (*n* = 49), CALR (*n* = 34), *SRSF2* (*n* = 16), *TET2* (*n* = 10), *U2AF1* (*n* = 11), *EZH2* (*n* = 7), MPL (*n* = 6), *IDH2* (*n* = 5), *IDH1* (*n* = 4), and *CBL* (*n* = 1). The cumulative incidence of non-relapse mortality (NRM) was 21% at 1 year and 25% at 5 years. The 5-year rates of PFS and OS were 48% and 56%, respectively. In a multivariate analysis, the CALR mutation was an independent factor for lower NRMs (*p* = 0.05), improved PFS (*p* = 0.01), and OS (*p* = 0.03). *ASXL1* and *IDH2* mutations were independent risk factors for a lower PFS (*p* = 0.008) and OS (*p* = 0.002), whereas no impact was observed for “triple negative” patients [16]. Table 2 summarizes the additional mutations described in patients with PMF and their potential pejorative prognostic impact on OS and/or AML transformation.

Besides conventional diagnostic and/or prognostic approaches, it is quite difficult to propose a therapeutic strategy based on only one or two genes. For instance, in a multi-institutional collaborative project, 1473 patients with MPNs were screened for only *IDH1* and *IDH2* mutations: 594 with ET, 421 with PV, 312 with PV, 95 with post-PV/ET MF, and 51 with blast-phase MPN. A total of 38 IDH mutations were detected in 5 (0.8%) patients with ET, 8 (1.9%) patients with PV, 13 (4.2%) patients with PMF, 1 (1%) patient with post-PV/ET MF, and 11 (21.6%) patients with blast-phase MPN (*p* < 0.01). Mutant *IDH* was documented in the presence or absence of MPN driver mutations with similar mutational frequencies (1% in patients with PV and patients with ET and 4% in patients with PMF). We can note a larger frequency of the *IDH* mutation occurrence in patients that are nullizygous for the *JAK2* haplotype, especially in PMF, and at a lower risk for a present complex karyotype when secondary leukemia occurs. In chronic-phase PMF, *JAK2* haplotype nullizygosity but not *IDH* mutational status had an adverse effect on OS. In contrast, in both patients with blast-phase PMF and patients with blast-phase PV/ET, the *IDH* mutation predicted a worse survival [11]. Although interesting and rigorous, it is not possible to make a therapeutic decision (in particular, allo-HSCT) based only on these results without data on the other HRMs and the patients’ clinical histories. In the same way for blast-phase MPNs as described by Venton et al. in 2017, conventional clinical factors (age, karyotype, ELN2017 prognostic classification, treatments received, treatment response, and allo-SCT) failed to predict the patients’ outcomes. Only the mutational status accessed via NGS appeared relevant to predict the patients’ prognoses at the blast phase. Three genes, *TP53* (*p* = 0.001), SRSF2 (*p* = 0.018), and *TET2* (*p* = 0.012), impacted AML prognosis pejoratively at the transformation time. In 33% of cases, the mutation profile was already present for MPNs in the chronic phase. However, only the patients with a *SRSF2* mutation presented a lower LFS (52 months) vs. unmutated patients (134 months) (*p* < 0.001) [65]. Based on these results, it seems, nevertheless, difficult to propose allo-HSCT as a first-line treatment for young/fit patients with chronic-phase MPNs and with the *SRSF2* mutation.

## 4. Conclusions

The use of somatic myeloid NGS panels in routine biology plays an increased role in MPN management, especially as a precious diagnostic aid for triple-negative MPNs, but also, and more importantly, as a method to stratify the risk profile of young/fit patients with PMF. However, first-line allo-HSCT is not systematically proposed for young/fit patients with PMF and HRMs at diagnosis because of the high risk for NRMs of this therapy. Further comparative and prospective studies are needed to better access the impact of a more aggressive treatment on PFS and OS in patients with PMF and HRMs at diagnosis. In contrast, performing NGS in patients with ET/PV and a *JAK2/CALR/MPL* driver mutation or patients with PMF who are ineligible for allo-HSCT does not present optimal cost-efficiency, since these data do not have a significant impact on our clinical practice.

## Figures and Tables

**Figure 1 cells-12-00105-f001:**
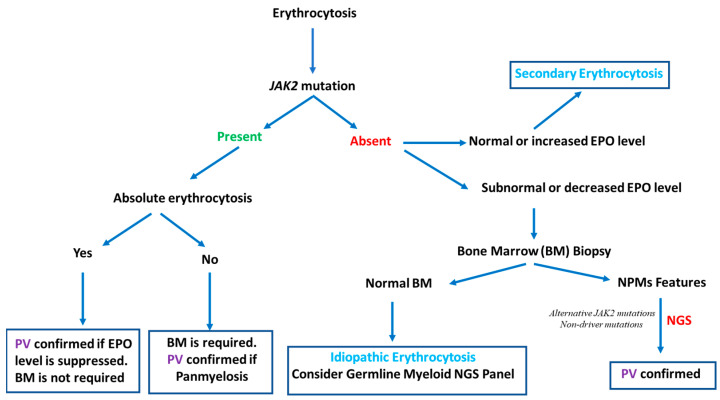
Erythrocytosis and PV diagnostic approaches: NGS contribution to PV diagnosis.

**Figure 2 cells-12-00105-f002:**
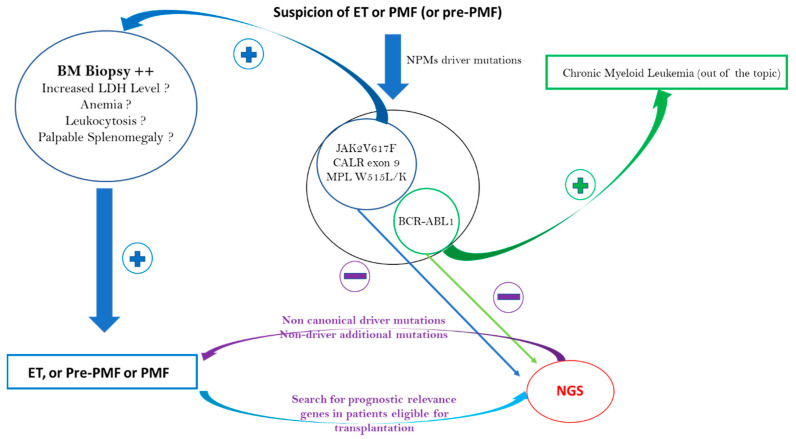
ET, pre-PMF, and PMF diagnostic approaches: NGS contribution to diagnosis.

**Table 1 cells-12-00105-t001:** Recurrent additional mutations in patients with MPN.

Involved Function	Gene	Frequency (%) PV/ET/PMF	Pathogenic Mechanism	References
DNA Methylation	*TET2*	15%/10%/15–30%	Loss of function of *TET2*	[7,8,9]
			Expansion of differentiation	
			Mechanism unknown	
	*DNMT3A*	3–7%/6–7%/5–10%	Loss of function of *DNMT3A*	[7,9,10]
			Blocking differentiation mechanism unclear via interaction with polycomb repressive complex 1	
	*IDH 1/2*	1%/1%/2–5%	Gain of function of IDH1/2	[11,12]
			Modification of metabolism of αKG via conversion on the oncometabolite, D-2-hydroxyglutarate, and blocking cellular differentiation	
Histone Modification	*ASXL1*	3–10%/4–10%/20–40%	Mechanism unknown: blocking methylation of H3K4 and H3K27 induces aberrant gene expression	[9,13,14,15,16,17]
	*EZH2*	1–3%/1%/5%	Mechanism unknown: prevents methylation of H3K27 and increases expression of *EZH2* in tumoral cells	[18,19]
mRNA Splicing	*SF3B1*	1–2%/2–3%/10%	Modification of HEAT domain of *SF3B1* induces aberrant splicing	[9,20]
	*SRSF2*	1%/1%/7–18%	Hotspot P95 induces aberrant differential splicing of many genes via gain of function and increasing affinity	[10,13,15,21,22]
	*U2AF1*	<0.5%/1%/10-15%	Hotspots at S34 and Q157 modify zinc finger and affect splicing through RNA binding activity, but they have different effects on 3′ splice-site recognition	[7,23]
	*ZRSR2*	1–2%/1–2%/10%	Loss of function of *ZRSR2* mutations causes abnormal splicing via intron retention of U12-dependent minor introns	[7,10]
Signaling	*LNK/SH2B*	1–3%/0–5%/0–6%	Lost negative feedback regulation of TPO and MPL signaling and increase in activation of JAK/STAT signalization	[7,13,24,25,26]
	*CBL*	1%/1–2%/5%	Gain function: affects E3 ubiquitin ligase and activation of LYN signal, PI3K, and reduces apoptosis cells	[7,10,27,28]
	*NRAS/KRAS*	1%/1%/3–4%	Gain function: constitutive activation of *RAS* proteins	[7,10,28,29]
	*PTPN11*	<0.5%/<0.5%/1%	Unknown	[10]
Transcription Factors	*RUNX1*	1%/1%/1–4%	*RUNX1* mutation induces alteration of Wnt signalization and abrogated p53 function	[30]
	*NFE2*	2–3%/1%/1–5%	Unknown	[31,32]
	*PPM1D*	2%/1%/1%	Mechanism unclear: malignancy by suppression of DNA repair, cell cycle arrest, and apoptosis	[10,33]
	*TP53*	1%/1%/1–4%	*TP53* plays a central role in regulating cellular responses to genotoxic stress, and loss of *TP53* provides a selective advantage for neoplastic growth	[7,10]

**Table 2 cells-12-00105-t002:** Prognostic impact of additional mutations described in PMF.

Gene	Frequency in PMF (%)	Pejorative Impact on OS	Increase in the AML Transformation Risk	References
** *ASXL1* **	25	**Yes**	**Yes**	[13,17,58]
** *TET2* **	10.3	No	No	[59]
** *SF3B1* **	9	No	No	[60]
** *SRSF2* **	7.5	**Yes**	**Yes**	[61]
** *CBL* **	6	No	No	[62]
** *DNMT3A* **	5.5	No	**Yes**	[63]
** *TP53* **	4	**Yes**	No	[64]
** *EZH2* **	3.3	**Yes**	No	[14]
** *IDH1-2* **	1.5	**Yes**	**Yes**	[65]

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
