# Peer review of "Impact of Molecular Biology in Diagnosis, Prognosis, and Therapeutic Management of BCR::ABL1-Negative Myeloproliferative Neoplasm"

_cells, 2022, doi:10.3390/cells12010105_

Round 1

Reviewer 1 Report (New Reviewer)

Abbou et al reviewed the molecular genetics in Ph-neg MPN.

Although the article deserves merit as it addresses an important topic and debated issues within the subject, indeed, the article needs to be edited for English language, as in the present version it has so many typos that make really difficult even to understand the meaning in some paragraphs.

Major issues

Please carefully check typos (NPMs instead of MPNs on line 44 and many other lines, MFP instead of PMF on line 48; there is a “on” missing on line 47; differenciation instead of differentiation in Table 1; correct “a aberrante” into “an aberrant”; “lead determining” line 53-54, please rephrase in a more intelligible sentence and much more)

Minor issues

The updated nomenclature should be used preferentially for gene rearrangements (BCR::ABL1 instead of BCR-ABL1)

The names of genes should be reported in italics

Author Response

Although the article deserves merit as it addresses an important topic and debated issues within the subject, indeed, the article needs to be edited for English language, as in the present version it has so many typos that make really difficult even to understand the meaning in some paragraphs.

Thank you for your remarks.

The manuscript has undergone complete English revision by MDPI.

Please find attached the MDPI English Editing Certificate and the revised manuscript.

Major issues

Please carefully check typos (NPMs instead of MPNs on line 44 and many other lines, MFP instead of PMF on line 48; there is a “on” missing on line 47; differenciation instead of differentiation in Table 1; correct “a aberrante” into “an aberrant”; “lead determining” line 53-54, please rephrase in a more intelligible sentence and much more)

Thanks you for your comments.

  • NPMs has been changed for MPNs line 44 and in all other lines in the revised manuscript
  • The manuscript has undergone complete English revision by MDPI.
  • MFP has been changed for PMF in line 48 and in all other lines

Minor issues

The updated nomenclature should be used preferentially for gene rearrangements (BCR::ABL1 instead of BCR-ABL1)

The updated nomenclature is now used in the revised manuscript. Thank you for your comment.

The names of genes should be reported in italics

The name of genes is now reported in italics in the revised manuscript.

Reviewer 2 Report (New Reviewer)

The review entitled “Impact of molecular biology in diagnosis, prognosis, and therapeutic management of BCR-ABL1-negative Myeloproliferative Neoplasms” describes the biological, clinical and prognostic characteristics of the BCR-ABL1-negative Myeloroliferative Neoplasms such as Polycythemia Vera, Essential Thrombocythemia, Primary Myelofibrosis. I think that the descriptive approach is well-structured and the authors wrote this review in didactic mode. This aspect provides a easy reading and understandable. Based on these considerations, although the review article is  par excellence is decoid of originality, I think that this manuscript is suitable for publication in its current version.

Author Response

Thank you for your remark and your comments.

Round 2

Reviewer 1 Report (New Reviewer)

The authors have addressed the raised issues, I have no further comments

This manuscript is a resubmission of an earlier submission. The following is a list of the peer review reports and author responses from that submission.

Round 1

Reviewer 1 Report

Dear Authors,

I am sorry to reject the manuscript that I have extensively reviewed. While the topic may be of interest, there are several points of concern, in terms of both style and content.

Below you can find the major issues:

- The work language is poor and presents several shortfalls that make both the reading and the understanding difficult.

- The  abstract reproduces exactly some passages of the manuscript and should be reformulated.

- As far as content, some of the studies discussed don't use NGS as a platform. Some studies discussed in detail are already obsolete and/or incorporated in later works, especially in the third paragraph.

- The MIPSS+ is presented as a prognostic score in the transplant context, whereas it encompass more or less all the studies presented in the paragraph and has been validated as a very robust prognostic score in any MF context. 

- The GIPSS score is not discussed at all. 

- The "to go further" paragraph does not present any novelty and can be easily incorporated in the previous one.

Author Response

Dear reviewer,

Thank you for taking the time to read this review and propose significant ameliorations and/or corrections.

I will try to answer any queries, point by point, you've addressed.

  • The work language is poor and presents several shortfalls that make both the reading and the understanding difficult.

In the final and corrected version of the paper, English will be checked by a native English-speaking colleague.

  • The  abstract reproduces exactly some passages of the manuscript and should be reformulated.

Thank you for your comment. Abstract will be modified, to meet your expectations, in the final manuscript.

  • As far as content, some of the studies discussed don't use NGS as a platform. Some studies discussed in detail are already obsolete and/or incorporated in later works, especially in the third paragraph.

Thank you for these comments. Regarding the work of Guglielmeli et al, notably, you are right, it doesn’t use NGS as plateform.. It is confusing if one refers to the title. That’s why, for clarity, this reference and the related paragraph will be removed from the revised manuscript.    

  • The MIPSS+ is presented as a prognostic score in the transplant context, whereas it encompass more or less all the studies presented in the paragraph and has been validated as a very robust prognostic score in any MF context.

I understand your comment. This paragraph will be rewritten based in this remark in the revised manuscript. Thank you.  

  • The GIPSS score is not discussed at all. 

Thank you for this constructive remark. GIPSS score is now discussed from the line 238 to the line 247of the revised manuscript.

  • The "to go further" paragraph does not present any novelty and can be easily incorporated in the previous one

Thank you for your comment. The paragraph “to go further” is now removed and incorporated in the previous one

Reviewer 2 Report

The review is accurate and comprehensive. Actually it points out the lack of sufficiently significant prognostic data to indicate the use of NGS in ameliorating the selection of patients for transplantation, independently from other scores (DIPSS, DIPSS+), and in driving the choice of therapies, due to the lack of target therapies, with the exception of IDH inhibitors. There are two minor misspellings: line 116 diver, driver; line 182 Guglielmeli, Guglielmelli.

Author Response

Dear reviewer,

Thank you for taking the time to read this review and your considerate comments.

  • The misspellings: line 116 diver, driver; is now corrected in the revised manuscript.
  • Based on the comment of another reviewer, this paragraph based on the work of Guglielmelli has been removed

Thank you again for you time.

Reviewer 3 Report

This is a well written, comprehensive and well referenced review of the role of NGS in MPNs. The only issue I have is that the topic has been covered reasonably well by other groups. These papers have been referenced and there is additional valuable discussion points in this review. I think it is worthy of publication in Cells. The type in Figure 2 is a little difficult to read. I suggest a different non-italicised font with increased font size would help

Author Response

Dear reviewer,

Thank you for taking the time to read this review and your considerate comments.

  • Concerning your first point: The only issue I have is that the topic has been covered reasonably well by other groups.

You are completely right. That’s why, on the one hand, I tried to bring my personal touch to this work. And, on the other hand, I always have scrupulously cited each reference (original articles and/or previous reviews) from which I was inspired. It's quite hard to reinvent everything.

  • The type in Figure 2 is a little difficult to read. I suggest a different non-italicised font with increased font size would help

Thank you for your remark. I have changed for a non-italicised and larger type in the revised manuscript.

Thank you again for you time.